# Hierarchical Constraints on the Distribution of Attention in Dynamic Displays

**DOI:** 10.3390/bs14050401

**Published:** 2024-05-11

**Authors:** Haokui Xu, Jifan Zhou, Mowei Shen

**Affiliations:** Department of Psychology and Behavior Sciences, Zhejiang University, Hangzhou 310023, China; jifanzhou@zju.edu.cn

**Keywords:** attention, visual motion perception, hierarchical representation, cueing effect

## Abstract

Human vision is remarkably good at recovering the latent hierarchical structure of dynamic scenes. Here, we explore how visual attention operates with this hierarchical motion representation. The way in which attention responds to surface physical features has been extensively explored. However, we know little about how the distribution of attention can be distorted by the latent hierarchical structure. To explore this topic, we conducted two experiments to investigate the relationship between minimal graph distance (MGD), one key factor in hierarchical representation, and attentional distribution. In Experiment 1, we constructed three hierarchical structures consisting of two moving objects with different MGDs. In Experiment 2, we generated three moving objects from one hierarchy to eliminate the influence of different structures. Attention was probed by the classic congruent–incongruent cueing paradigm. Our results show that the cueing effect is significantly smaller when the MGD between two objects is shorter, which suggests that attention is not evenly distributed across multiple moving objects but distorted by their latent hierarchical structure. As neither the latent structure nor the graph distance was part of the explicit task, our results also imply that both the construction of hierarchical representation and the attention to that representation are spontaneous and automatic.

## 1. Introduction

Our daily experiences are immersed in environments bustling with numerous moving objects that rarely exist in isolation. Instead, these objects often interrelate, forming complex structures, such as a flock of geese in flight. Recent research highlights the proficiency of the visual system in extracting shared motion characteristics across multiple objects, thereby forming hierarchical representations of motion [1,2,3]. This capability enables the visual system to efficiently interpret and predict dynamic scenes with limited resources.

Some classic visual motion effects already imply the existence of hierarchical motion representations. For instance, the Duncker Wheel phenomenon illustrates how the observed motion of a wheel’s rim, akin to pendulum movement, transforms into the perception of a rotating wheel when combined with its pivot’s horizontal motion (a circular motion superimposed on a horizontal motion) [4]. Similar effects were observed in other physical motions and biological motions [5,6]. These phenomena reveal our ability to perceive the structure underlying motion, even in stimuli with two or three moving dots. Further evidence comes from computational modeling studies in recent years. Gershman introduced a Bayesian vector analysis theory explaining the visual system’s approach to hierarchical motion perception [1]. This involves decomposing motion into common relative movements and employing Bayesian inference to resolve ambiguities in vector decomposition, favoring the hierarchical structure with the highest posterior probability. Computational models simulating this process have shown remarkable efficacy in elucidating classical visual motion phenomena. Subsequent research confirms the visual system’s reliance on Bayesian reverse engineering to discern hierarchical structures across diverse motion scenarios, including circular motion, complex physical motion, and social interaction motion, enhancing our ability to perceive, understand, and predict motion [2,7,8,9]. This foundational understanding is crucial for effective interaction with our surroundings and social partners.

The extensive investigation into visual hierarchical representation has significantly advanced our understanding of motion processing. However, it remains unclear that how this advantage is achieved, that is, how other aspects of the vision are affected by hierarchical representations. As a pivotal aspect of vision research, attention represents an information selection mechanism, channeling limited resources to specific information segments [10,11,12]. Objects within the focus of attention are more accurately tracked and remembered [13,14], and they assume greater importance in decision-making processes [15,16,17]. Therefore, it would be worthwhile to explore how visual attention operates with a hierarchical motion representation.

The influence of an object’s spatial structure on attentional distribution has been a focal point of research, revealing that attention can swiftly and automatically spread among visual objects and perceptual grouping [18,19]. Specifically, transitioning attention from one spatial point to another typically incurs a delay. However, this delay diminishes when the points are part of the same object or group. Recent findings extend this phenomenon to social groups, demonstrating accelerated attention shifts within groups perceived as collaborative entities, such as two discs in pursuit of a target or social interactions signified by handshakes [20,21]. These studies collectively suggest that attentional distribution across multiple objects is not merely a function of spatial proximity but is intricately linked to our mental representation of these objects. The mental closeness of the objects facilitates quicker attention transitions. Therefore, it is plausible to assume that visual motion hierarchical representation similarly influences attentional distribution, enhancing the speed of attention shifts between motion objects that are closer in the hierarchical framework.

In order to investigate the proposed hypothesis, the present study aims to manipulate the distances between moving objects within their latent hierarchical representations and assess the resultant shifts in attention. In a hierarchical structure, the distance between two objects can be quantified as the “minimum graph distance” (MGD). It is defined as the minimum number of edges required to go from one object to another in the hierarchy. Manipulation of MGD can be achieved by constructing hierarchies with varying numbers of edges or by designing a fixed hierarchy with objects positioned such that the shortest paths between any two objects differ. We will adopt these two approaches to manipulate MGD in two experiments, respectively, and generate motion corresponding to the specific hierarchical structure by the hierarchical model. To measure attentional shifts, we will use the classic congruent–incongruent cueing paradigm [18]. In this paradigm, cue and target appear sequentially on either the same object (valid condition) or two different objects (invalid condition). The cue’s high likelihood of validity encourages participants’ attention to focus on the cued object. Under the invalid condition, attention redirects to the object where the target is located, resulting in a longer reaction time. The cueing effect, a measure of attention, is determined by the difference in reaction times between valid and invalid conditions. Smaller cueing effects suggest quicker attentional shifts from the cued to the target object. In sum, through two experiments, this study will analyze attention transfer between moving objects within different (Experiment 1) and fixed (Experiment 2) hierarchical structures, aiming to elucidate the influence of latent hierarchical representations on attentional distribution.

## 2. Experiment 1

### 2.1. Method

#### 2.1.1. Participants

A priori power analysis was conducted with the G*Power 3.1. We were mainly concerned with two within-subject factors, MGDs (0/1/2) and cue types (valid/invalid). Since there were no previous similar studies, we chose the effect size considered to be the criterion for median effects ($\eta_{p}^{2}=0.138$) and α=0.05. A sample size of 18 participants was sufficient to detect an effect with a power of 0.95. A total of 18 undergraduate and gradate students (12 females and 6 males, aged 18 to 26 years) at Zhejiang University participated in this experiment for credit or monetary payment. One participant was excluded from further analysis due to an equipment error resulting in loss of data. All participants had normal or corrected-to-normal vision and signed informed consent. The study was approved by the Institutional Review Board at the Department of Psychology and Behavioral Sciences of Zhejiang University.

#### 2.1.2. Apparatus and Stimuli

The stimuli and program were created using MATLAB R2021b and Psychtoolbox 3.0 and presented on a 17-inch CRT monitor with a spatial resolution of 1024 × 768 pixels at a 100 Hz refresh rate. Participants maintained a visual distance of approximately 70 cm from the center of the monitors.

All stimuli were presented on a grey screen (size: 36.6° × 27.6°; RGB = [150, 150, 150]). Each object was a solid red disk with diameter of 1.0° (RGB = [150, 0, 0]). Motion displays containing 2 objects were generated using a hierarchical generation model (Figure 1b). At any time, each node of the hierarchical tree was assigned a motion vector according to the Gaussian process. All motion vectors on the path from the object’s child node to the root node were superimposed, and Gaussian noise was added to obtain the motion of the object. As a result, the root node shared by all objects contained the general direction of motion, and since each object had different child nodes, there was still some difference in their motion. The speed of motion was set at 0.12°/frame, and the directions were changed by an average of 7.05°/frame.

#### 2.1.3. Procedures and Design

Attention was probed by the classic congruent–incongruent cueing paradigm. Each trial began with a cross presented as a fixation point in the center of the screen. After 0.5 s, the fixation point disappeared, and two discs appeared and started moving. The movement lasted for 3 s and then stopped. At the same time, one of the small discs briefly appeared white (RGB) as a cue. The cue disappeared after 100 ms, and after another 200 ms, the target letter (T or L) was presented on one of the discs, and the distractor was presented on the other disc (Figure 1a). Participants were required to recognize the identity of the target letter and respond by pressing the corresponding key (T or L on the keyboard) within 2 s. Once the key was pressed, all stimuli disappeared from the screen, and feedback was given. The next trial started after an interval of 1–1.75 s.

In order to manipulate the MGD of objects in the hierarchical structure, we designed three potential hierarchical structures (Figure 1c). In the same-node structure, two objects shared the same root node, and their apparent motion differences were mainly caused by different superimposed noises. Substituting the motion vector in the root node of the example in Figure 1b, both objects moved to the right. At this time, the MGD of these two objects in the hierarchical structure is 0 steps. In the same-branch structure, the two objects also shared the same root node, but one object had a different child node. Substituting the motion vector of the example in Figure 1b without the child node of object 1, object 1 moved to the right and object 2 moved in a diagonal downwards motion. The MGD of two objects in the layer structure was 1 step. In the diff-branch structure, in addition to sharing the same root node, the two objects each had a different child node. Substituting the motion vector of the example in Figure 1b, object 1 moved in a diagonal upwards motion, while object 2 moved in a diagonal downwards motion. The MGD of these two objects in the hierarchical structure was 2 steps. An example of motion animation can be found at https://osf.io/8p23q/?view_only=28d4c25ee43e4e6aa264fc8ef7e6574f (accessed on 23 April 2024). In order to eliminate the interference of spatial distance in attention, we kept the average spatial distance of the two objects and the spatial distance of the final stopping moment the same in the motion-generating process. Therefore, the difference in the distribution of visual attention on moving objects corresponding to three different hierarchical structures can be reasonably attributed to the differences caused by the latent hierarchical structure. There were two types of cues in the congruent–incongruent cueing paradigm. When the cue was valid, the target would appear on the object where the cue was located. Conversely, when the cue was invalid, the target was presented on another object. At this time, participants needed to transfer their attention from the cueing object to the target object, leading to a slowing down of reaction time. The difference in response times between the invalid cue condition and the valid cue condition was the cueing effect, which can be used to measure attention. In order to ensure the attention was focused on the cueing object when the cue was valid, the probability of the valid cue should be much greater than the invalid cue. In this study, we set the probability of a valid cue at 2/3 and the probability of an invalid cue at 1/3. The experiment finally included 180 trials, with 60 trials corresponding to each hierarchical structure, of which 40 trials were valid cue conditions and 20 trials were invalid cue conditions.

#### 2.1.4. Data Analysis

We recorded and analyzed participants’ response time (RT) during the task. Trials with correct responses were included in the data analysis. We used repeated-measures ANOVAs and paired *t*-tests (two-tailed) for all statistical analyses. We reported the Greenhouse–Geisser-corrected *p*-value in ANOVAs if sphericity was violated. Additionally, Bonferroni correction was used in further comparisons.

### 2.2. Results

We first conducted a 3 (MGD) × 2 (cue type) repeated-measures ANOVA. The results revealed a significant main effect of MGD (*F*(2, 32) = 9.43, *p* < 0.001, $\eta_{p}^{2}$ = 0.371) and a significant main effect of cue type (*F*(1, 16) = 1065.34, *p* < 0.001, $\eta_{p}^{2}$ = 0.985). The interaction was also significant (*F*(2, 32) = 50.23, *p* < 0.001, $\eta_{p}^{2}$ = 0.758). Further comparison revealed that in all MGD conditions, RTs were longer when the cue was invalid, indicating that the setting of cue probability was effective (Figure 2).

The significant interaction effect suggested differences in RT latencies due to cue invalidation between MGDs. To directly compare the delay in RT, we obtained the cueing effect by subtracting the RT in the valid cue condition from the RT in the invalid cue condition and conducted a one-way repeated-measures ANOVA. The significant main effect of MGD (*F*(2, 32) = 50.12, *p* < 0.001, $\eta_{p}^{2}$ = 0.758) indicated that cueing effects were different among MGDs. Further comparison revealed that the cueing effect in the MGD with 1 step (same-branch) was significantly longer (*p* < 0.001, Cohen’s d = 1.541) than that in the MGD with 0 steps (same-node). The cueing effect in MGD with 2 steps (diff-branch) was significantly longer than that in MGD with 0 steps (*p* < 0.001, Cohen’s d = 2.038) and the MGD with 1 step (*p* = 0.003, Cohen’s d = 0.989). These results showed that the longer the MGD between the objects in the hierarchical structure, the more slowly the attention transferred, suggesting that the latent hierarchical structure limited the distribution of attention to multiple moving objects.

## 3. Experiment 2

The different levels of MGD in Experiment 1 originated from different hierarchical structures, which may introduce other possibilities for interpretation of the results. The reason for the difference in attentional distribution may be differences in other features between hierarchical structure (e.g., depth, total number of edges) or differences in motion trajectories. The purpose of Experiment 2 was to better control these factors and accurately detect the impact of MGD on attentional distribution.

### 3.1. Method

#### 3.1.1. Participants

The sample size was same as that in Experiment 1. A new group of 18 undergraduate and gradate students (8 females and 10 males, aged 19 to 29 years) at Zhejiang University participated in this experiment for credit or monetary payment. All participants had normal or corrected-to-normal vision and signed informed consent. The study was approved by the Institutional Review Board at the Department of Psychology and Behavioral Sciences of Zhejiang University.

#### 3.1.2. Apparatus and Stimuli

The apparatus and stimuli used in this experiment were the same as those in Experiment 1, with the exception of the elements introduced below. Motion displays containing 3 objects were generated using a hierarchical generation model (Figure 3a). They all share the same root node, and object 1 and object 2 share one same intermediate node. The MGD between object 1 and object 2 is shorter (2 steps), and the MGD from object 3 is the same and longer (3 steps). An example of generation process in Experiment 2 can be found at https://osf.io/8p23q/?view_only=28d4c25ee43e4e6aa264fc8ef7e6574f (accessed on 23 April 2024).

#### 3.1.3. Procedures and Design

The task was identical to that in Experiment 1. There were three objects in the moving displays, and the attention transfer caused by invalid cues could occur between any two objects (regardless of the direction of the transfer). There were three situations: when attention was transferred between object 1 and object 2, the MGD was 2 steps; when attention was transferred between object 1 and object 3, and between object 2 and object 3, the MGDs were both 3 steps. We planned to conduct two sets of comparisons. The first set was to compare the cueing effect between object 1 and object 3 and the cueing effect between object 1 and object 2. Such comparisons would help to demonstrate that a longer MGD will lead to slower attention shift times. The second set was used to compare the cueing effect between object 1 and object 3 and the cueing effect between object 2 and object 3. Such comparisons would help illustrate that when MGD was the same, the time taken to transfer attention was the same. We finally introduced 4 levels to the condition of MGD. The two levels of the first set of comparisons were called Diff_13 and Diff_12, respectively, and the two levels of the second set of comparisons were called Same_13 and Same_23, respectively.

It should be noted that for each set of comparisons, we needed to keep the spatial distance between objects at the end of the movement the same. Specifically, in the first set of comparisons, at the end of the motion, the spatial distance between object 1 and object 3 should equal that between object 1 and object 2. In the second set of comparisons, at the end of the motion, the spatial distance between object 1 and object 3 should equal that between object 2 and object 3. However, it was difficult to make the spatial distances between the three objects the same (in the shape of an equilateral triangle). Therefore, the motion trajectories of Diff_13 and Same_13 were not the same and could not be regarded as the same condition.

The probability of a valid cue was set as 2/3 and the probability of an invalid cue as 1/3. The experiment finally included 240 trials, with 60 trials corresponding to each MGD level, of which 40 trials were valid cue conditions and 20 trials were invalid cue conditions.

### 3.2. Results

We first conducted a 4 (MGD) × 2 (cue type) repeated-measures ANOVA. The results revealed a significant main effect of MGD (*F*(3, 51) = 8.92, *p* < 0.001, $\eta_{p}^{2}$ = 0.344) and a significant main effect of cue type (*F*(1, 17) = 282.99, *p* < 0.001, $\eta_{p}^{2}$ = 0.943). Further comparison revealed that in all MGD conditions, RTs were longer when the cue was invalid, indicating that the setting of cue probability was effective. The interaction was also significant (*F*(3, 51) = 10.239, *p* < 0.001, $\eta_{p}^{2}$ = 0.376), suggesting differences in RT latencies due to cue invalidation between MGDs (Figure 3b).

To directly compare the delay in RT, we obtained the cueing effect by subtracting the RT in the valid cue condition from the RT in the invalid cue condition and conducted a one-way repeated-measures ANOVA. The significant main effect of MGD (*F*(3, 51) = 10.311, *p* < 0.001, $\eta_{p}^{2}$ = 0.378) indicated that cueing effects were different among MGDs. Further comparison revealed that cueing effect in Diff_13 was significantly longer than that in Diff_12 (*p* < 0.001, Cohen’s d = 1.29), which indicated that attention shift slower when the MGD was longer. Cueing effects between Same_13 and Same_23 did not show significant difference (*p* = 0.639, Cohen’s d = 0.40), which implied that attention transfer took the same time when MGDs (and other features like spatial distance) were the same. These results again demonstrated that the latent hierarchical structure constrained the distribution of attention on multiple moving objects.

## 4. Discussion

Visual motion objects are not perceived in isolation but are represented as hierarchical structures. Revealing how vision processes information based on hierarchical structures is crucial for understanding human visual intelligence [22]. The current study explores how the latent hierarchical structure of moving objects influences the distribution of visual attention, employing a cueing paradigm. Experiment 1 demonstrated that attention transition is quicker for shorter minimum graph distances (MGDs) within varying hierarchical structures. Similarly, Experiment 2 revealed that within a constant hierarchical structure, reduced MGDs facilitate faster attention shifts. These findings indicated that the latent hierarchical representation of moving objects constrains the distribution of attention, implying that hierarchical representation could be an important basis for attention selection and further processing of motion information such as understanding and prediction.

Attentional distribution in dynamic scenes is markedly influenced by the spatial and motion information itself, which may interfere with, amplify, or mask the effect from latent hierarchical structures [23,24,25]. The current study meticulously ensured that the observed increase in the cueing effect was attributable to expanded MGDs within hierarchical representations through several strategies. Firstly, we controlled several basic motion features such as average speed and average steering angle in the generated motion trajectories, keeping them consistent, and maintained equal spatial distances between objects during both cue and target phases. Thus, differences in the RTs cannot be attributed to differences in these motion attributes. Moreover, participants may have preferences or be more sensitive to certain motion trajectories. These higher-level factors may lead to faster responses under specific motion trajectories and may affect the decision stage. However, Experiment 1‘s findings revealed consistent RTs under valid cue conditions across various hierarchical structures, indicating that motion trajectories did not directly influence responses. The alteration in the cueing effect predominantly stemmed from RT changes under invalid cue conditions, signifying that attention shifts were primarily responsible for RT differences. Additionally, motion information with different hierarchical structures always shows differences in the details of specific motion, so in Experiment 1, it was impossible to make the motion trajectories correspond exactly to different hierarchical structures. Experiment 2 eliminated such influence by fixing motion trajectories and latent hierarchical structures. Therefore, the differences in cueing effects can only be attributed to differences in MGDs. Furthermore, our analysis extended beyond comparing cueing effects across different MGDs, also verifying their consistency when the MGD remained constant. These results provided robust evidence from two perspectives, strongly advocating that the distance between objects in hierarchical representation constrains the distribution of attention. It should be noted that this study does not disregard the importance of spatial distance. The effect of spatial distance on attentional distribution has been well supported by research, which is why we kept the spatial distance the same. Our main point is that in addition to spatial distance, attentional distribution can also be affected by distance in representations. Since the attentional shift in the invalid cue condition is considered to rely on top-down control [26,27,28], our study provides a preliminary exploration of the role of hierarchical representation in top-down attention. Given that conditions with different spatial distances were not involved in the current study, further research is required to explore the interactive relationship between spatial distance and representational distance and how they jointly affect attentional distribution.

Research paradigms investigating hierarchical representation generally fall into two categories. One set of tasks explicitly measure the latent hierarchical structure, for example, asking participants to report the most likely structure [16] or to describe the scene [7,9]. Another set of tasks do not require participants to directly report; these tasks may involve change detection [29,30], tracking [3,31], or prediction of motion [2,32]. However, in these tasks, participants may benefit from hierarchical structures, since task performance could be improved through the construction and application of hierarchical representations. This improvement might motivate participants to actively infer latent hierarchical structures. Unlike previous studies, our experiment did not necessitate that participants explicitly report on or employ hierarchical structures for task completion. In fact, hierarchies did not enhance task performance and could potentially delay RTs. Nonetheless, the significance of the hierarchical structure emerged in our findings, providing the first evidence for the spontaneous construction of hierarchical representations. Furthermore, our results show that the constraints on attention by hierarchical representations are also spontaneous and automatic, aligning with existing studies on attentional distribution. Studies have demonstrated that visual objects, virtual contours, perceptual grouping, and social grouping impose similar constraints on attentional distribution, facilitating faster shifts within the same representation compared to between different representations [19,20,21,33]. Representing constraints on the distribution of attention possesses significant functional implications. When visual objects are part of the same representation, it indicates a reduced psychological distance or a closer relationship. Thus, the rapid transfer of attention between these objects aids in the identification, comprehension, and prediction of visual scenes. Our research contributes new insights regarding hierarchical representation, broadening our understanding of visual representation’s impact on attention. It highlights that not only does belonging to a common representation influence attentional distribution, but variations in the representation’s specific structure also affect it. Hierarchical representation not only offers a more detailed, multi-level description of the relationships between objects but also serves as a causal structure encompassing the generative processes of visual scenes [2,34,35,36]. The constraints of the allocation of attention through hierarchical representation enhance the visual system’s ability to comprehend the causal relationships between visual objects, thus capturing the essence of the visual scene more accurately. This, in turn, supports additional cognitive processes, including decision making and language comprehension [35,37,38].

Hierarchical representations play a crucial role in various vision tasks; for instance, objects within a hierarchical structure that share a common node are tracked more effectively, and neighborhoods closer to a target in the hierarchy aid significantly in location prediction [2,3]. Yet, the operational principles of the visual system utilizing hierarchical representations require further investigation. This research aims to undertake a preliminary exploration from the perspective of attention, a cognitive process essential for human intelligence, enabling selective information processing in dynamic environments [14,39]. The findings suggest that differential access to attentional resources might account for variances in tracking and prediction performance. When multiple objects share nodes in the hierarchical representation, their MGD in the hierarchy is shorter. This facilitates easier sharing of attentional resources, enhancing tracking capabilities. Similarly, attention shifts between them faster, making it easier to predict one’s motion based on the other’s motion. Admittedly, in addition to MGD, hierarchical structures encompass various distinct features, such as depth, direction, and causality. How these features influence attention and other aspects of vision are important topics for both psychology and neuroscience and need to be further explored in future studies.

## 5. Conclusions

In summary, the current study found that the latent hierarchical representation of dynamic scenes imposes constraints on attentional distribution. The closer the object is in the hierarchical representation (shorter MGD), the faster attention shifts between objects.

## Figures and Tables

**Figure 1 behavsci-14-00401-f001:**
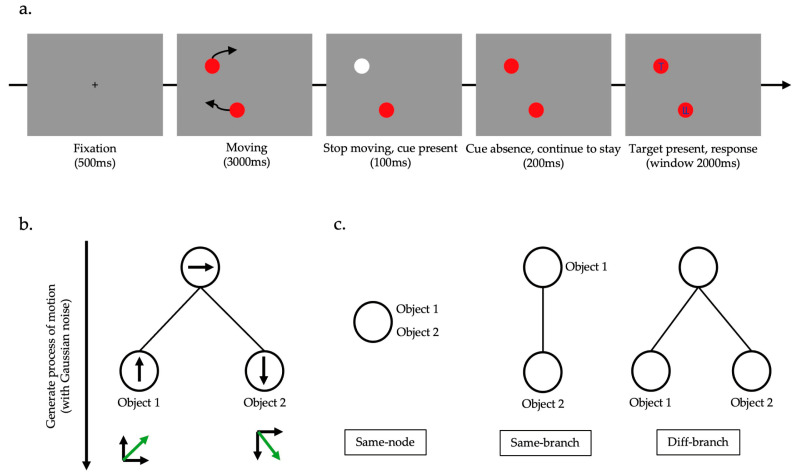
Procedure and design of Experiment 1. (**a**) The task used in Experiment 1. The white circle indicated the cue. Target letter would be “T” or “L” (in the illustration here is “T”), and the distractor is a combination of “T” and “L”. (**b**) Illustration of how the motion was produced by the hierarchical generation model. (**c**) The hierarchical structure used in Experiment 1.

**Figure 2 behavsci-14-00401-f002:**
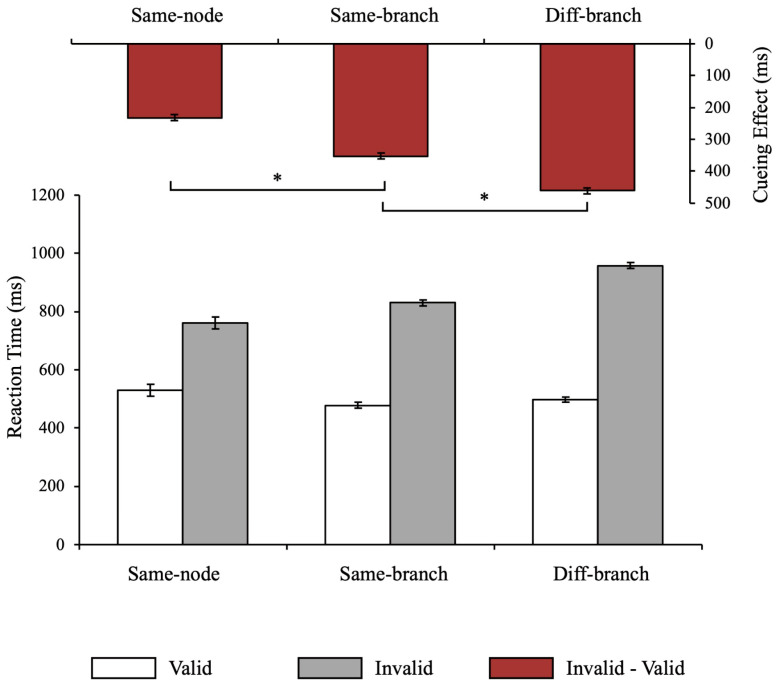
Results of Experiment 1. The histogram containing white and gray columns at the bottom is the reaction time under different conditions, and the histogram containing red columns at the top is the cueing effect under different conditions. Error bars represent SEM. Asterisks (*) indicate *p*-values of statistical test < 0.05.

**Figure 3 behavsci-14-00401-f003:**
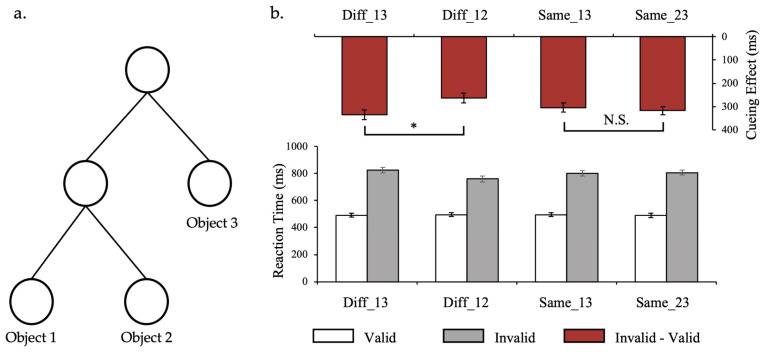
Hierarchical structure and results of Experiment 2. (**a**) The hierarchical structure with 3 objects used in Experiment 2. MGD between object 1 and object 2 was 2, and the MGD between object 1 and object 3 was identical to that between object 2 and object 3, being 3. (**b**) Results of Experiment 3. The histogram containing white and gray columns at the bottom is the reaction time under different conditions, and the histogram containing red columns at the top is the cueing effect under different conditions. Error bars represent SEM. Asterisks (*) indicate *p*-values of statistical tests < 0.05. N.S is the abbreviation of non-significant.

## Data Availability

Data are available from the corresponding author on reasonable request.

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
