# Peer review of "Hierarchical Constraints on the Distribution of Attention in Dynamic Displays"

_behavsci, 2024, doi:10.3390/bs14050401_

Round 1

Reviewer 1 Report

Comments and Suggestions for Authors

This research delves into the distribution of visual attention within a hierarchical motion framework. The manuscript is well-written; my only suggestion is to review the abstract for any potential typos. Additionally, considering the dynamic nature and latent hierarchical structure of the presentation stimuli, it could enhance reader comprehension if the authors provide online access to these stimuli.

Author Response

Thank you very much for your review and comments on the manuscript. We have corrected typos in the new manuscript and provided a link contained the motion animations for easier online access by reviewers and readers.

Reviewer 2 Report

Comments and Suggestions for Authors

1. What an incredibly tight and clear paper - this was absolute fun to read. The logic was clear throughout. The experiments were simple but elegant and well-designed. The General Discussion was also very convincing, and I think this paper would be a wonderful contribution to the literature. (I especially loved the point about how this work could help us understand how we predict an object's motion through the motion of a different object!)

2a. Perhaps my biggest point of improvement would be just to clarify the methods. In particular, I'm quite familiar with the Duncker Wheel and Gershman's work, but I think I still missed what a 'node' constitutes in the MGD. So on p.3-4 of the manuscript, I think it would be helpful to clarify this. My understanding is that the root node was a general direction of motion, and then object 1 could move in a diagonal upwards motion, while object 2 could move in a diagonal downwards motion. Does this sound about right? If so, I would include this in the manuscript. Perhaps this can be done in Section 2.1.2 when Figure 1b is referenced. 

2b. And then in Section 2.1.3, when the different conditions are being laid out, I think it would be helpful to be explicit / concrete about what the different conditions actually translated to in terms of the motions. So for example, taking Figure 1b: in the same node case, both objects would be moving to the left; in the same branch case, both objects would be moving to the left and up (??); and in the diff branch case, one object would be moving diagonal up and the other, diagonal down. (In any case, Figure 1c is not referenced in the paper, so it might be helpful to slow down and just describe this more concretely too.)

2c. For the same reason as above, I struggled a little to understand what the motions could have looked like in Experiment 2. If we go one more level down the tree, what other dimension of motion did they use for the next level? Further clarification in this section would be helpful. (Same as above, perhaps be more concrete and describe Figure 3a?)

Minor comments:

1. In line 9, "oerate" should be "operate".

2. In line 279, "spacial" should be "spatial".

3. In the abstract, I would consider noting that Experiment 2 was used to control for other potential confounds - because I found myself confused about what "asymmetric hierarchy" meant, and what this would add to the overall story.

4. Could a link to the motion animations be made available, especially for the same-branch vs. diff-branch cases, and the three-level hierarchical motion case? Given that this paper's key contribution is the psychophysical exploration of how attention operates over hierarchical motion patterns, I think having the demos readily available would help for comprehension.

END OF REVIEW

Author Response

  1. Thank you very much for your review and compliments on the manuscript. The suggestions you provided were very helpful in improving the quality of the manuscript.
  2. We have added descriptions of how the hierarchical structure generates motion in the Methods section. We also concreted about how the different conditions actually translated to in terms of the motions in the experiment and provided a link contained the motion animations for easier online access by reviewers and readers.
  3. Besides, thank you for pointing out the typos, which have been corrected in the new manuscript.

All the words and sentences changed in the new version of manuscript have been marked in red.